# Thermodynamic dissipation constrains metabolic versatility of unicellular growth

Tommaso Cossetto[1,2], Jonathan Rodenfels [2,3] ✉ & Pablo Sartori [1] ✉

Metabolic versatility enables unicellular organisms to grow in vastly different environments. Since growth occurs far from thermodynamic equilibrium, the second law of thermodynamics has long been believed to pose key constraints to life. Yet, such constraints remain largely unknown. Here, we integrate published data spanning decades of experiments on unicellular chemotrophic growth and compute the corresponding thermodynamic dissipation. Due to its span in chemical substrates and microbial species, this dataset samples the versatility of metabolism. We find two empirical thermodynamic rules: first, the amount of energy dissipation per unit of biomass grown is largely conserved across metabolic types and domains of life; second, aerobic respiration exhibits a trade-off between dissipation and growth, reflecting in its high thermodynamic efficiency. By relating these rules to the fundamental thermodynamic forces that drive and oppose growth, our results show that dissipation imposes tight constraints on metabolic versatility.

A fundamental characteristic of life is its ubiquity across diverse environments. At the core of this ubiquity lies cellular metabolism, an interconnected network of chemical reactions that transforms environmental nutrients into the mass and energy that support growth. Metabolism is highly versatile, which is manifested by life's spread in environments spanning a wide range of nutrient types, temperatures, and other physico-chemical parameters. Because the matter and energy fluxes sustained by metabolism keep cells far from thermodynamic equilibrium, it has long been speculated that the second law of thermodynamics poses essential constraints to life[1-5]. However, far from equilibrium there is no general relation between fluxes and the free energy that these fluxes dissipate. Therefore, despite remarkable efforts[6-13], little is known about the interplay between thermodynamics and metabolism. Are there general quantitative rules that constrain the versatility of metabolism by the free energy it dissipates?

In the last decades, studies of metabolism have mirrored molecular biology's great advances and focused on genome-scale metabolic networks[11,14-17]. In contrast, pioneering descriptions of metabolism dating as far back as a century ago were macroscopic[18], and summarized cellular growth by the substrates and products exchanged with the environment. This macroscopic approach identifies a metabolic type with a "macrochemical equation", e.g. $C_6H_{12}O_6 + O_2 + NH_3 \rightarrow$ biomass + $CO_2$ + $H_2O$ for glucose respiration, and is sufficient for a thermodynamic description of steady unicellular growth[19-21] (see Boxes 1–3 for a summary of the formalism). Metabolic versatility is then embodied in the large variety of electron donors and acceptors characterizing metabolic types (in the previous example, glucose and oxygen respectively), as well as in the perturbations to growth within each type[22-25]. Applications of this macroscopic approach in biotechnology[26-29] and ecology[30-32] were aimed at predicting biomass yield, yet hinted to a fundamental relationship between thermodynamics and microbial metabolism. However, metabolic versatility arises from both, evolutionary adaptations (gain or loss of metabolic pathways) as well as plastic response to environmental changes (activation of alternative pathways). Therefore, exploring the thermodynamic constraints to metabolic versatility necessitates experiments on cellular growth far beyond what a single research group can achieve.

In this work, we apply the formalism of non-equilibrium thermodynamics to published measurements spanning eight decades of research on unicellular growth. This results in a database that consists of 504 instances of growth experiments under defined conditions with well-characterized non-equilibrium thermodynamics (Supplementary Information B). Analysis of this database reveals two empirical rules that link the versatility of metabolism to thermodynamic dissipation.

[1]Gulbenkian Institute for Molecular Medicine, Oeiras, Portugal. [2]The Max Planck Institute of Molecular Cell Biology and Genetics, Dresden, Germany. [3]Cluster of Excellence Physics of Life, TU Dresden, Dresden, Germany. ✉e-mail: rodenfels@mpi-cbg.de; pablo.sartori@gimm.pt

## Results

### Non-equilibrium thermodynamics of unicellular growth

The framework we used to analyze the data allows to compute thermodynamic dissipation from the fluxes of chemicals exchanged between a growing population of cells and its environment. Box 1 summarizes this approach, which is further elaborated in Supplementary Information C.

While this framework is applicable to experimental data in which all exchange fluxes are measured, in practice most of the available data lacks this level of detail. Often only a few fluxes are measured, leaving the remaining fluxes undetermined. Moreover, it is common that published works report normalized fluxes, called yields, instead of fluxes, thus removing the characteristic timescale of the problem. To overcome the first limitation in the data, we inferred the undetermined yields from element conservation, a procedure akin to balancing the stoichiometry of a chemical reaction. Concerning the lack of timescale, we use normalized thermodynamic parameters, which cancels their timescale. In Box 2 we elaborate these two points, whereas in Supplementary Fig. 8 we characterize a subset of chemostated data for which thermodynamic parameters can be inferred per unit time. Finally, we remark that because the dissipation is an explicit function of the fluxes, we expect these quantities to exhibit correlations. Throughout this paper, we will disentangle the nature of these correlations on the database.

An example of how the formalism described above can be applied to experimental data is provided in Box 3.

### Dataset validation

The dataset analyzed in this work covers more than one hundred metabolic types belonging to all domains of chemotrophic life, vastly expanding previous approaches[28,30,32]. Furthermore, it incorporates over 70 species of microbes and a variety of physico-chemical conditions for each type, thus constituting an ideal resource to investigate the interplay between thermodynamics and metabolic versatility. To asses the quality of this dataset, we validated three key physical principles: element conservation, the first law of thermodynamics, and the second law of thermodynamics.

Element conservation was used in this work to estimate yields that were not measured (Box 2 and Supplementary Information C3). To

validate this approach, we compared the estimated yields $y_i$ of both, substrates and products per electron donor, with direct measurements of those same yields whenever such additional measurements were available. Figure 1a shows a scatter plot of the 69 different yields from 56 experiments for which such validation was possible (see also Supplementary Fig. 1 and Supplementary Information J). There is a good agreement, with the largest deviations corresponding to small product yields, which are generally challenging to measure accurately.

The first law of thermodynamics establishes the equality between enthalpy and heat exchanges (Box 1). Therefore, we can validate our thermodynamic approach by comparing the predicted net exchange of enthalpy with direct calorimetric measurements of heat, both quantities normalized per amount of biomass grown. Figure 1b shows this comparison for the subset of data (98 experiments) that contained calorimetric measurements (see also Supplementary Fig. 1 and Supplementary Information J). As before, there is a general agreement between the predicted and measured heat exchanged per unit biomass synthesized.

Finally, we verified the second law of thermodynamics. For each instance of our dataset we computed the free energy dissipated per electron donor consumed (see Box 2). We found that in more than 99% of the data there was a net dissipation of free energy, in agreement with the second law. We discarded the remaining data points from further analysis.

Having established the quality of the dataset, we proceed to investigate the interplay between physiological and thermodynamic growth parameters.

### Variability in yield and dissipation highlights metabolic versatility

A key physiological parameter of growth is the biomass yield, $y$, which quantifies the amount of biomass produced per electron donor consumed. The biomass is expressed in units of carbon moles (Cmol), which is the amount of dry biomass containing one mole of carbon atoms (see Box 2 as well as Supplementary Information G2, C2 and I). We observe wide variability in $y$ throughout the dataset (Fig. 2a). Yield variability spans three orders of magnitude across metabolic types. We remind the reader that a metabolic type is defined as a particular set of substrates and products of metabolism, e.g. glucose respiration is a metabolic type, and glucose fermentation is a different type (full list in Supplementary Fig. 2). In contrast to the large variability in yield across types, the variability within types spans under a factor of two. This within type variability reflects changes in experimental conditions (such as temperature, dilution rate, or pH), unicellular species, and experimental uncertainty. The lowest yield in the dataset, $y \sim 10^{-2}$ Cmol mol$^{-1}$, corresponds to autotrophic methanogenesis, an ancient archeal metabolic type that synthesizes biomass and methane from $CO_2$ and molecular hydrogen in anoxic environments[34–36]. The highest yields, $y \approx 10$ Cmol mol$^{-1}$, are produced by the respiration of disaccharides, such as sucrose and lactose, reaching values close to their carbon limit of 12 Cmol mol$^{-1}$ (the number of carbon atoms present in the electron donor).

To account for the thermodynamics of growth, we quantify the free energy dissipation per electron donor consumed, $\sigma$ (hereafter simply referred to as dissipation, Box 2 and Supplementary Information C). As in the case of yields, we find that $\sigma$ varies substantially, also spanning three orders of magnitude, (Fig. 2b). It is similarly bounded by autotrophic methanogenesis, $\sigma \approx 50$ kJ mol$^{-1}$, which synthesises biomass from the most oxidized form of carbon, $CO_2$[37]; and respiration of sugars, $\sigma \approx 6$ MJ mol$^{-1}$, which are highly reduced electron donors. In contrast, no clear distinction in $y$ nor $\sigma$ was found between archaea, prokarya, and eukarya beyond the distinction due to changes in metabolic type (Supplementary Fig. 3). Similarly, temperature and pH had no systematic effect on $y$ (Supplementary Fig. 4). Thus, the data demonstrates that variability in yield and dissipation are dominated by changes across metabolic types.

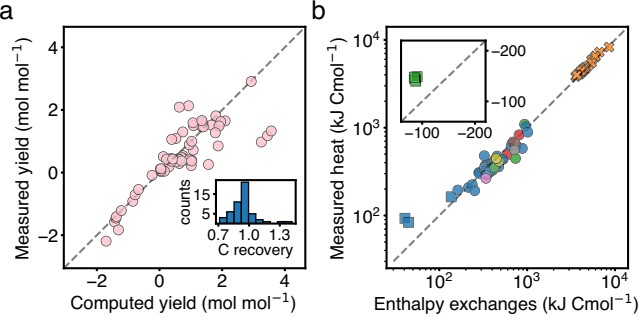

**Fig. 1 | Thermodynamic validation of the dataset. a** Scatter plot of measured yields against yields computed by element conservation (69 data points). Yields are given per unit of electron donor, and in the plot are negative for substrates and positive for products. Dashed line is a reference, indicating a perfect agreement. Inset. Results of 51 measurements of carbon recovery, i.e. global carbon balance of the experiment, as reported in the original references. A value of one corresponds to a full recovery of the carbon metabolized from substrates to products and biomass. **b** Comparison of the balance of enthalpy exchanges with the heat measured per biomass synthesized (both quantities are multiplied by a minus sign to be plotted on a log-scale) for 98 data points. The first law of thermodynamics equates the two quantities, Eq. (2) and dashed line, and is approximately satisfied. Inset. Examples of endothermic growth data, where heat is absorbed.

## BOX 1
# Thermodynamics of cellular growth in a nutshell

Consider a population of cells growing at a constant rate $\gamma$. Substrates ($i \in$ sb) are imported at fluxes $f_i^+ > 0$ (in units of moles per volume of biomass and time) and transformed into biomass and products. At steady state, the composition of biomass as well as its density $\rho_C$ (in units of carbon moles per volume) are constant (Supplementary Information B). Products ($i \in$ pd) are exported at fluxes $-f_i^- < 0$.

This non-equilibrium process is constrained by the second law of thermodynamics[7,33]:

$$\dot{s}_{prod} = -\frac{\dot{q}}{T} - \sum_{i \in sb} f_i^+ s_i + \sum_{i \in pd} f_i^- s_i + \gamma s \geq 0 \qquad (1)$$

where $\dot{s}_{prod}$ is the rate at which entropy is produced, $\dot{q}$ the rate at which heat is produced (or absorbed, if positive), $T$ the temperature, $s_i$ the entropy of the $i$-th chemical, and $s$ the biomass entropy per unit volume of biomass (Supplementary Information I).

Energy conservation during growth is expressed by the first law of thermodynamics[7,33]:

$$\dot{q} = -\sum_{i \in sb} f_i^+ h_i + \sum_{i \in pd} f_i^- h_i + \gamma h, \qquad (2)$$

where $h$ is the biomass enthalpy (Supplementary Information I) and $h_i$ the enthalpy of the corresponding chemical species. This equivalence between heat and enthalpy is also known as Hess' law.

Putting together the expressions of the first and second law, and using the definition of chemical potential, $\mu_i \equiv h_i - Ts_i$, we arrive at

$$T\dot{s}_{prod} = \sum_{i \in sb} f_i^+ \mu_i - \sum_{i \in pd} f_i^- \mu_i - \gamma g \geq 0, \qquad (3)$$

with $T\dot{s}_{prod} = -\dot{g}_{diss}$. This equation decomposes the total rate of free energy dissipation, $\dot{g}_{diss} \leq 0$, in terms of free energy influx of substrates, $i \in$ sb, free energy outflux of products, $i \in$ pd, and free energy flux due to biomass synthesis, $\gamma g$, where $g$ is the free energy of biomass (Supplementary Information I). Taken together, the above constitutes a minimal non-equilibrium thermodynamic framework to describe cellular growth.

## The dissipative cost of growth is conserved across metabolic types

To investigate the relationship between thermodynamics and physiology of unicellular growth, we plot the dissipation against the yield (Fig. 3a, Supplementary Figs. 5, 6). Two salient features become apparent: first, yield and dissipation are strongly correlated over the span of three orders of magnitude; and second, data corresponding to the same metabolic type cluster together in quasi-linear spaces following distinctive trends. These two observations suggest separating the dissipation into a mean contribution specific to a particular metabolic type, $\bar{\sigma}$, and a small deviation due to within type variability, $\delta\sigma$, and analogously for the yield, $\bar{y}$ and $\delta y$ (Fig. 3a inset). To ease notation, we omit the explicit dependence of these variables on the type (Supplementary Information D).

Plotting the mean dissipation against the mean yield across types, we find that they are linearly proportional (Fig. 3b). This correlation persists when the axis are expressed per gram or per carbon mole, albeit over a smaller range (Supplementary Fig. 7). Therefore, we define the coefficient

$$\alpha \equiv \bar{\sigma}/\bar{y}, \qquad (6)$$

which quantifies the dissipative cost of growth for a certain metabolic type. As before, we do not write the explicit dependence of $\alpha$ on the type. This cost, which by the second law must be (and is) positive, measures the loss of free energy per biomass grown. The linear relation with small intercept implies that $\alpha$ is conserved across different metabolic types. The cost $\alpha$ has a median value $\approx 500$ kJ Cmol$^{-1}$ or 8 ATP equivalents per carbon atom fixed (Table 1, Supplementary Information E), and an interquartile range of 350 kJ Cmol$^{-1}$ (see Fig. 4 and corresponding section for a discussion on variability). This value of $\alpha$ is consistent with a previously hypothesized constant used to predict biomass yields in the bioengineering literature[28]. However, it is

larger than biophysical estimates[38,39] (Supplementary Information E), suggesting that bookkeeping of intracellular processes leaves out significant contributions to dissipation. The cost $\alpha$ is conserved irrespective of the species, the domain of life, the growth rate, or other physico-chemical variables (Supplementary Figs. 8 and 4), presenting an overall constraint to metabolic versatility.

## Deviations in yield and dissipation within metabolic types

Within a given type, the deviation $\delta\sigma$ relates linearly to $\delta y$. This is a direct consequence of element conservation, which was used to compute the dissipation from measured yields (Box 2, Box 3, and Supplementary Information C3). In contrast to the single empirical trend observed between $\bar{\sigma}$ and $\bar{y}$ across types, the slope relating $\delta\sigma$ to $\delta y$ in each type does vary substantially across types. Aerobic types have a steep negative slope while anaerobic types have a shallow negative slope (Fig. 3c and Supplementary Fig. 9). To quantify the diversity of trends we define a coefficient, $\beta$, by the following linear regression within a metabolic type,

$$\delta\sigma \approx \beta\delta y. \qquad (7)$$

The coefficient $\beta$ accounts for how variations in dissipation are adjusted relative to variations in yield within a type. We find that in all instances $\beta < 0$, and thus increases in yield are accompanied by a reduction in dissipation. For anaerobic types we find $\beta \approx -100$ kJ Cmol$^{-1}$, and thus the yield can be increased with little effect on the dissipation (Fig. 3c, dashed line). Instead, aerobic types show $\beta \approx -500$ kJ Cmol$^{-1}$, and thus there is a significant trade-off between yield and dissipation: relative increases in yield of 50% result in comparable reduction in dissipation (Fig. 3c, dotted line). In summary, we find that while the dissipative cost of growth, quantified by $\alpha$, is conserved across metabolic types, the deviations within type, quantified by $\beta$, distinguish aerobic from anaerobic types.

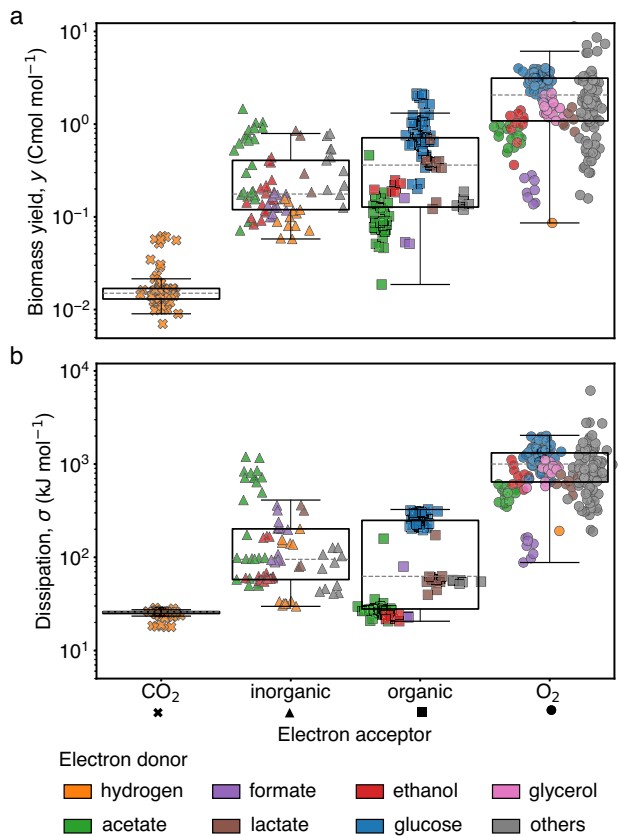

**Fig. 2 | Variability of yield and dissipation in unicellular growth. a** The biomass yield $y$, biomass produced (in Cmol) per electron donor consumed (in mol), spans across three orders of magnitude. The data includes over one hundred different metabolic types (each defined by the set of substrates and products utilized, full list in Supplementary Fig. 2), which we grouped in the figures by class of electron acceptor (marker) and electron donor (color for most frequents). This same color-marker scheme is preserved throughout the paper. In the plots, the boxes delimit the first and third quartiles, the dashed lines represent the median and the whiskers are drawn at the farthest data point within 1.5 times the interquartile range from the box. We use the same convention throughout the paper. **b** The dissipation $\sigma$, free energy dissipated per electron donor consumed, displays a variability that spans a range comparable to that of yield. The box plots are defined as in panel (**a**). In both panels, the boxes from left to right describe $n = 71, 85, 109, 239$ data points, respectively.

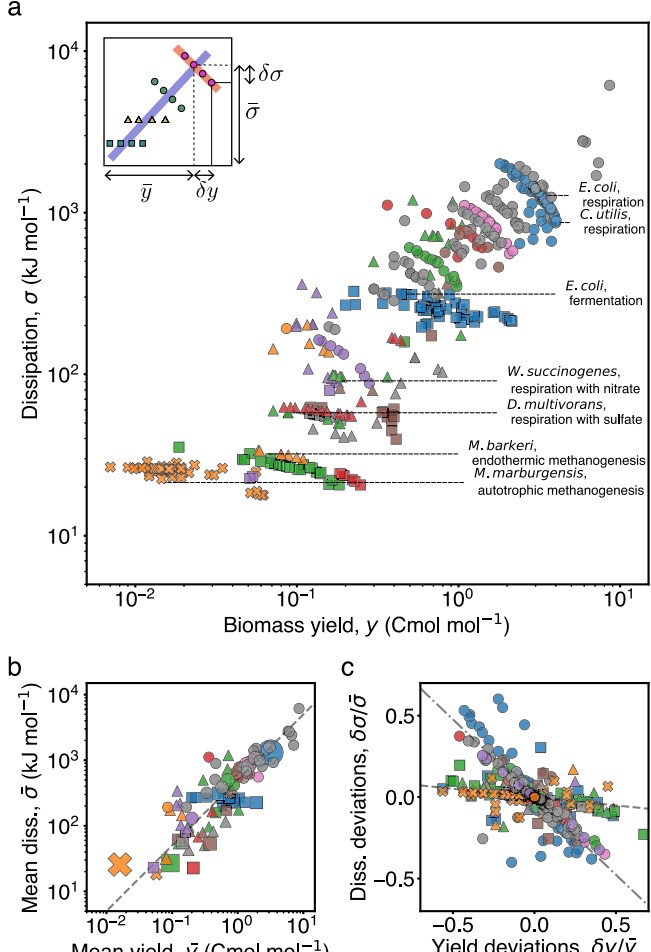

**Fig. 3 | Dissipation and yield are related across and within metabolic types.**
**a** Scatter plot of the dissipation, $\sigma$, versus the yield, $y$, for the whole dataset. The variability across types results in a strong correlation of the data over three orders of magnitude (see Supplementary Fig. 7 for different units). Data from the same type cluster together, in quasi-linear spaces. Inset depicts the notions of average dissipation/yield and corresponding deviations. **b** Mean dissipation, $\bar{\sigma}$, plotted against yield, $\bar{y}$, for each type. The marker size represents the amount of data in each type. The dashed line in the background corresponds to, $\bar{\sigma} = \alpha\bar{y}$, fixing $\alpha = 500\,\text{kJ}\,\text{Cmol}^{-1}$, which is approximately the median across types and approximates well the data. **c** Normalized deviations in dissipation, $\delta\sigma/\bar{\sigma}$, are plotted against normalized deviations in yield, $\delta y/\bar{y}$, for each metabolic type. Dashed and dotted lines correspond to linear regressions for anaerobic and aerobic types with slopes $-0.10 \pm 0.01$ and $-0.95 \pm 0.03$, respectively.

## Cost is conserved despite variability of free energy exchanges

To assess the extent of conservation in the cost, we decomposed it into three terms: free energy absorbed with substrates; released with products; and stored in biomass (Fig. 4a and Box 2). Each of these terms provides a reference for cost variability.

Across metabolic types, the free energy absorbed per biomass produced is high and variable, with a median value of 900 kJ Cmol$^{-1}$ and an interquartile range of 2700 kJ Cmol$^{-1}$ (Fig. 4a, second box). Similarly, the free energy released with the products is also high and variable, with a median value of 1500 kJ Cmol$^{-1}$ and an interquartile range of 2700 kJ Cmol$^{-1}$ (Fig. 4a, third box). In contrast, the free energy content of biomass is roughly one order of magnitude lower (Fig. 4a fourth box and Supplementary Information I2). We conclude that the cost $\alpha$ is dominated by the free energy exchanged through substrates and products. Remarkably, while these exchanges are large and variable, $\alpha$ is lower and exhibits a comparatively low variability: median 470 kJ Cmol$^{-1}$ and interquartile range 350 kJ Cmol$^{-1}$ (Fig. 4a first box, and inset for histograms). Therefore, the cost is conserved over a wide range of free energies imported and exported (Fig. 4b, c).

The reason for the low variability in $\alpha$ is that, while the free energies absorbed and released are both large and variable, they are strongly correlated and compensate each other (Fig. 4d). Such compensation arises from an interplay between element conservation and the free energies of chemicals, which we disentangle in the following sections. In summary, while the free energy flowing through cells varies significantly across metabolic types, the compensation of the influx and outflux of free energy results in a conserved dissipative cost of growth.

## Thermodynamic implications of the anaerobic/aerobic transition

Due to the versatility of metabolism, many electron donors can be metabolized both aerobically and anaerobically. To investigate the thermodynamics of this phenomenon, we grouped metabolic types based on their electron donor and acceptor, and identified 30 pairs of

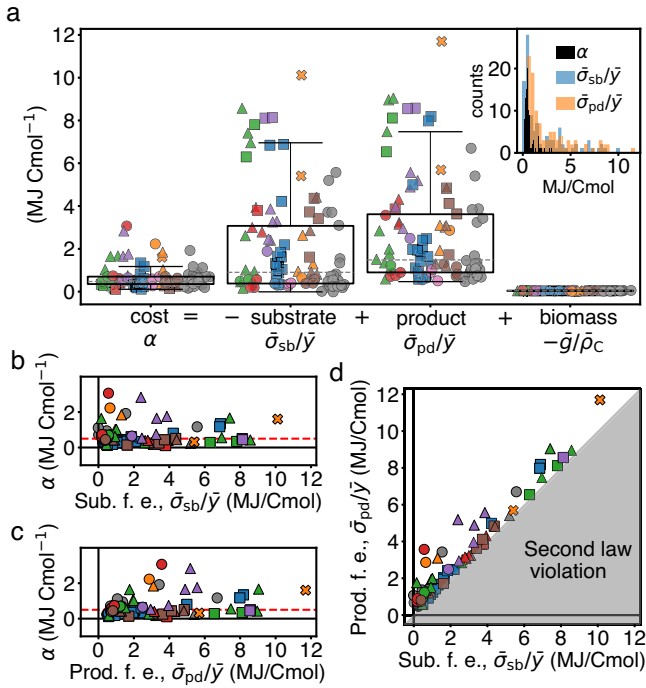

**Fig. 4 | Decomposition of the dissipative cost $\alpha$. a** The dissipative cost, $\alpha = -\bar{\sigma}_{sb}/\bar{y} + \bar{\sigma}_{pd}/\bar{y} - \bar{g}/\bar{\rho}_C$, is decomposed into the free energy lost due to substrate consumption, $-\bar{\sigma}_{sb}/\bar{y}$, the free energy created due to product formation, $-\bar{\sigma}_{pd}/\bar{y}$, and the free energy stored into biomass, $\bar{g}/\bar{\rho}_C$, all per unit biomass synthesized (see Box 2 for the definitions, overbars denote means within types, Supplementary Information D). Notice that the free energies of formation of biomass, $\bar{g}$, and of most chemical species are negative, making $\bar{\sigma}_{sb,pd} > 0$. The contribution from free energy storage in biomass is small and its variability is not visible (Supplementary Information I2). Substrate and product free energy fluxes are large and variable, yet compensate each other to result in a comparatively small and conserved cost of biomass. The box plots are defined as in Fig. 2a. Each box describes $n = 102$ data points. Inset. Histograms of $\alpha$, $\bar{\sigma}_{sb}/\bar{y}$, and $\bar{\sigma}_{pd}/\bar{y}$, showing skewness and small variability of $\alpha$. The cost as a function of the influx of free energy of substrates (**b**), and of the outflux of free energy of products (**c**). Compared to the large variability in the free energy influx and outflux, the cost is largely conserved. The red dashed line corresponds to $\alpha = 500$ kJCmol⁻¹. **d** Free energy outflux of metabolic products vs free energy influx of substrates. The data covers only a small linear space within the physically realizable region (unshaded area), indicating that the cost of growth is much smaller than influx/outflux of free energy. The shaded area corresponds to violation of the second law, which requires $\alpha > 0$ (a small offset due to biomass free energy is not visible).

groups that use the same electron donor both aerobically and anaerobically. For example, one pair consists of a group of metabolic types respiring ethanol with oxygen, and of a group of types respiring ethanol with sulfate. Aerobic groups are exclusively respiratory, using oxygen as electron acceptor, while anaerobic groups include both, fermentation as well as anaerobic respiration, and thus use a variety of electron acceptors other than oxygen. We find that, while in each pair yield and dissipation are generally higher for aerobic groups (Fig. 5a, b), the change in cost across this transition exhibits no clear trend (Fig. 5c). Furthermore, this is true when the transition corresponds to different species (e.g. *Candida utilis* respiring ethanol with oxygen or *Desulfococcus propionicus* respiring ethanol with sulfate), as well as when it corresponds to plastic changes of the same species (e.g. *E. coli* respiring or fermenting glucose). Therefore, irrespective of whether it is evolutionary or plastic, a transition from an anaerobic to an aerobic metabolic type that preserves electron donor increases yield and dissipation proportionally, satisfying the constraint of conserved cost, $\alpha$.

Instead, the thermodynamic signature of such transitions lies in how dissipation is adjusted relative to yield within metabolic type. This

is described by the coefficient $\beta$, the slope of the linear relation between dissipation and yield arising from element conservation. Figure 5e–g displays scatter plots of the dissipation and yield deviations for three anaerobic/aerobic pairs exemplifying the general rule, and Fig. 5h for a pair representing an exception (Supplementary Fig. 9 shows all the deviations within metabolic types). As one can see for the general case, aerobic types exhibit a marked trade-off between dissipation and yield, absent for the same electron donor in non-aerobic types. This difference in $\beta$ between anaerobic and aerobic types extends to almost all pairs (Fig. 5d). Therefore, a transition from anaerobic to aerobic metabolism preserving the same electron donor allows to trade dissipation for yield.

We identified a few outliers to this rule: metabolic types performing anaerobic respiration with electron acceptors such as nitrate, among few others (Fig. 5d, h). Those outliers harbor a $\beta$ similar to the value of aerobic types. Overall, we conclude that the thermodynamic implication of a transition from an anaerobic to an aerobic type lies predominantly in the emergence of a dissipation-yield trade-off.

## Opposing thermodynamic forces control dissipation and efficiency

So far we have established that the dissipative cost of growth is conserved across types, and that a trade-off between dissipation and yield deviations accompanies aerobic respiration. We now characterize these two metabolic signatures following the formalism of thermodynamics far from equilibrium. First, we note that cellular dissipation is given by a sum of products between yields, $y_i$, and their corresponding free energies, $\mu_i$ (Eq. (5)). However, as already anticipated, these yields are not independent within a metabolic type due to constraints imposed by element conservation (Eq. (4)).

To disentangle these constraints, we decompose the dissipation into independent contributions only. Each contribution is the product of an independent yield (e.g. biomass, electron donor, etc.) coupled to a generalized thermodynamic force $r_i$. The thermodynamic forces $r_i$ quantify the free energy difference driving the corresponding fluxes that sustain growth. Each force contains a linear combination of the chemical potentials of dependent substrates and products, with coefficients determined by the stoichiometry of the metabolic type[40]. Therefore, we can write the dissipation per electron donor as a function of the independent yields (Box 2 and Supplementary Information F1),

$$\sigma = r_{ed} - y\, r_b + \Delta, \tag{12}$$

where $\Delta = \sum_{i_i \in sb} y_{i_i} r_{i_i} - \sum_{i_i \in pd} y_{i_i} r_{i_i}$ and $i_i$ spans only the subset of independent yields of substrates and products other than biomass and electron donor. Within each metabolic type, the forces $r_i$ are the same, up to minor deviations due to biomass composition and temperature corrections (Supplementary Information F1, F2), but can change arbitrarily from type to type.

Roughly half of the metabolic types in the database have only one independent yield, the biomass yield $y$, and therefore $\Delta = 0$. Moreover, we found that for most of the remaining data $\Delta$ is negligible. Therefore, we identify two dominant thermodynamic forces of metabolism: the electron donor force, $r_{ed}$, coupled to the influx of electron donor molecules, and the biomass force, $-r_b$, that couples to biomass synthesis. The two forces $r_{ed}$ and $r_b$ can be heuristically associated with catabolism and anabolism, respectively (Fig. 6a).

We empirically find that $r_{ed}$ is positive and approximates well the mean dissipation, i.e. $\bar{\sigma} \approx r_{ed} > 0$ (Fig. 6b). As a consequence, $r_{ed}$ can be understood as the driving force of growth. The conserved cost $\alpha$ arises from a linear relationship across metabolic types between this force and the biomass yield, related to previous observations for "catabolic reactions"[26,29]. In contrast, we find that the anabolic force opposes dissipation, $-r_b < 0$, and due to element conservation is approximately

## BOX 2
# A framework for experimental data

A large fraction of the data we parsed reports yields instead of fluxes. Furthermore, often only the yields of few substrates or products were measured. Therefore, the framework of Box 1 needs to be adapted to this type of data.

The yield of a chemical $i$ is defined as the ratio of its flux to a reference flux, typically the electron donor, i.e. $y_i = f_i^{\pm}/f_{ed}^{+}$ (in units of mole per mole, by definition $y_{ed} = 1$). Similarly, the biomass yield is given by $y = \gamma \rho_C / f_{ed}^{+}$ (in units of carbon moles per mole). Supplementary Information C2 gives details on the definition of yield and Supplementary Information I1 on the choice of units.

To reconstruct the full set of yields from a subset of measured yields, we use element conservation[20,33]. Since the elements composing molecules are conserved by metabolic processes, the amount of a given element in the substrates metabolized must be equal to the amount secreted in products and synthesized in biomass (Supplementary Information C3 and F1). Therefore, the yields are constrained by an equation for each element ($k$ = C, H, O, N, ...):

$$\sum_{i \in sb} e_{i,k}\, y_i = \sum_{i \in pd} e_{i,k}\, y_i + e_{b,k}\, y, \tag{4}$$

where $e_{i,k}$ is the number of $k$ atoms in chemical $i$ and $e_{b,k} = \rho_k/\rho_C$ is the amount of element $k$ per carbon atom in biomass (by definition $e_{b,C} = 1$).

Note that Eq. (4) is completely equivalent to a "macrochemical equation".

Once all yields are determined, the free energy dissipated per electron donor consumed, $\sigma = -\dot{g}_{diss}/f_{ed}^{+}$ (referred as "dissipation" in the main text), can be expressed in terms of yields as

$$\sigma = \underbrace{\sum_{i \in sb} y_i \mu_i}_{-\sigma_{sb}} - \underbrace{\sum_{i \in pd} y_i \mu_i}_{-\sigma_{pd}} - y\, g/\rho_C \geq 0 \tag{5}$$

We have also defined the dissipation per electron donor by substrate import, $\sigma_{sb}$, and product export, $\sigma_{pd}$ (see Supplementary Information K for a summary table of symbols). The values of $\mu_i$ and $g/\rho_C$ correspond to free energies of formation that we extracted from public sources, see Supplementary Information H and I. Note that as these free energies are typically negative, we find that in most cases $\sigma_{sb}$ and $\sigma_{pd}$ are positive. Eq. (5) is the analogue of Eq. (3) using yields instead of fluxes, which results in canceling out the timescale of dissipation.

These adaptations of the framework of Box 1 allow to infer non-equilibrium thermodynamic properties from growth measurements, see also Box 3 for an example.

---

equal to the coefficient $\beta$, i.e. $\beta \approx -r_b$ (Fig. 6c and Supplementary Information F). As a consequence of its sign, the anabolic force can be understood as an opposing force to growth. The equivalence between $r_b$ and $\beta$ allows to trace the trade-off between dissipation and yield in aerobic metabolism to the large and negative free energy content of carbon dioxide (Supplementary Information F4). Therefore, dissipation is dominated by two opposing forces, catabolic and anabolic, confirming the notion that cell growth is governed by the principle of free energy transduction[41].

The decomposition of dissipation into independent terms, Eq. (12), further allows to define the thermodynamic efficiency of growth, $\eta$, following ref. 42 (see Supplementary Information F3). Intuitively, the efficiency measures the ratio of free energy produced and consumed by independent fluxes. In our case, the independent fluxes are largely dominated by biomass production and electron donor consumption, and so $\eta \approx -y\, r_b/r_{ed}$, which can be understood as a ratio of anabolic to catabolic dissipation. Figure 6d shows that aerobic types have a much larger efficiency than anaerobic types. This is because, due to the relationships established before, we have that $\eta \approx -\beta/\alpha$ (Fig. 6b, c shows the quality of this approximation), which together with the finding that $-\beta$ is larger for aerobic types results in their high efficiency. Therefore, despite aerobic biomass synthesis being opposed by a greater thermodynamic force, aerobic metabolism is more efficient. We remark that different definitions of growth efficiency have been proposed in the literature, see e.g. ref. 26. As defined here, $\eta$ takes explicitly into account stoichiometric constrains, and combines the empirical rules observed in this work into one quantity. Importantly, our definition is different from the common notion of ATP yield, which counts the net number of ATP molecules produced per electron donor catabolized (e.g. $\approx 32$ and 2 for glucose respiration and fermentation, respectively).

## Discussion

The metabolic versatility characterizing unicellular growth is thermodynamically constrained by two empirical rules. First, the non-equilibrium cost of biomass, $\alpha$, is largely conserved across adaptive and plastic changes in metabolism; and second, aerobic metabolism results in a trade-off between deviations in dissipation and yield, with coefficient $\beta$. The second rule concerns the value of $\beta$, the slope of a linear relation imposed by element conservation within a metabolic type. The first rule is an observation across types, and therefore not related to stoichiometric constraints. These rules are controlled by two separate thermodynamic forces, $r_{ed}$ and $r_b$, associated with catabolism and anabolism. The opposing signs of the catabolic and anabolic terms imply that cellular growth occurs through free energy transduction, akin to the transformation of chemical energy into mechanical work in molecular motors[41]. The decomposition into thermodynamic forces allows to define the efficiency of unicellular growth, dominated by the ratio of $\beta$ and $\alpha$, which is consistently higher for aerobic metabolic types.

Deviations from the two thermodynamic rules single out exceptional metabolic types. For example, hydrogenotrophic methanogenesis in archeans (orange crosses in Fig. 3) has a cost three-fold higher than the typical value[43], deviating from the first rule. Likewise, anaerobic respiration of nitrate can have an anabolic opposing force and thermodynamic efficiency comparable to the aerobic one, and thus deviates from the second rule. Interestingly, these metabolic types play a central role in theories of origin of life, eukaryogenesis and the evolution of aerobic metabolisms[22,36,44,45]. Finally, the existence of an also exceptional endothermic metabolic type (Fig. 1b Inset) proves that chemical entropy can be the dominant source of dissipation[46]. To what extent heat or chemical entropy dominates dissipation remains an important open question.

Our data collection and processing builds on decades of research on the energetics of microbial growth. Early studies were aimed at controlling microbial processes in bioreactors by predicting the biomass yield from electron donor properties, such as its degree of reduction[7]. Later meta-analysis suggested a more complex empirical relation between electron donor properties and the dissipative cost of growth[26]; a correlation between the free energy of the "catabolic reaction" and biomass yield[21,29]; and thermodynamic corrections due to non standard conditions[30]. Remarkably, it was also proposed that a constant value of

## BOX 3
# Extracting data from a paper: an example

Consider an experiment in which a population of *Escherichia coli* cells grows aerobically in batch culture. The media is minimal, composed of glucose ($C_6H_{12}O_6$), ammonia ($NH_3$), and micronutrients. Most of the data reports a few yields instead of all the fluxes needed to characterize cellular growth (Box 1 and 2). In this example, the experiment reports only a biomass yield of $y = 2.5$ Cmol/mol and a biomass composition of $e_{b,H} = 1.88$, $e_{b,O} = 0.44$, $e_{b,N} = 0.22$.

To complete the information on the yields of substrates and products, we can use element conservation, Eq. (4). This set of equations can be represented as a "macrochemical equation", which defines the metabolic type:

$$C_6H_{12}O_6 + y_{O_2}O_2 + y_{NH_3}NH_3 \rightarrow$$
$$y\,CH_{1.88}O_{0.44}N_{0.22} + y_{CO_2}CO_2 + y_{H_2O}H_2O \qquad (8)$$

where $y_{O_2}$ is the yield of oxygen per mole of glucose consumed, and the same for other chemicals. The unknown yields are obtained balancing the macrochemical equation[7,26,31], i.e. solving:

$$\underbrace{\begin{bmatrix} 6 & 0 & 0 & 1 & 1 & 0 \\ 12 & 0 & 3 & 1.88 & 0 & 2 \\ 6 & 2 & 0 & 0.44 & 2 & 1 \\ 0 & 0 & 1 & 0.22 & 0 & 0 \end{bmatrix}}_{e_{ik}} \underbrace{\begin{bmatrix} -1 \\ -y_{O_2} \\ -y_{NH_3} \\ 2.5 \\ y_{CO_2} \\ y_{H_2O} \end{bmatrix}}_{y_i} = 0. \qquad (9)$$

Once we have a value for all the yields $y_i$, we calculate the dissipation per electron donor, $\sigma$, using Eq. (5). To do so, we substitute $\mu_i$ with the free energy of formation of that chemical, corrected for temperature (Supplementary Information H). To estimate the free energy of formation of biomass, $g/\rho_C$, we use its element composition (Supplementary Information I). The heat produced per electron donor is computed analogously to the dissipation, but using enthalpies instead of free energies, which allows for comparison with calorimetric data.

We can now derive the fundamental forces governing the free energy transduction of the aerobic growth of *E. coli* on glucose. We use the independent yield, $y$, to express all other dependent yields. Restricting the matrix in Eq. (9) to dependent species $i_d$, we can invert it (entries $(e^{-1})_{i_d,k}$), to arrive at:

$$y_{CO_2} = E_{CO_2,\text{gluc}} - y\,E_{CO_2}, \qquad (10)$$

where $E_{CO_2,\text{gluc}} = \sum_k (e^{-1})_{CO_2,k}\,e_{\text{gluc},k}$ and $E_{CO_2} = \sum_k (e^{-1})_{CO_2,k}\,e_{b,k}$, and analogous expressions for the other chemicals. Using Eq. (10) in Eq. (5), we obtain $\sigma$ as a function of the biomass yield only,

$$\sigma = r_{\text{gluc}} - y\,r_b, \qquad (11)$$

where $r_{\text{gluc}} = \mu_{\text{gluc}} - \sum_{i_d} E_{\text{gluc},i_d}\,\mu_{i_d}$, and $r_b = g/\rho_C - \sum_{i_d} E_{i_d}\,\mu_{i_d}$. The coefficients $r_b$ and $r_{\text{gluc}}$, which have dimensions of a chemical potential, are the thermodynamic forces coupled to the independent yield in the flux-force expression of the dissipation[40]. In general, for cases with more than a single independent yield, the expression for $\sigma$ contains a term for each independent yield, Eq. (12).

Different experiments for the same metabolic type will report different biomass yields, due to e.g. changes in temperature or to experimental variation. This yield variability will be linearly related to the variability in dissipation by Eq. (11). Indeed, for every metabolic type, the forces $r_i$ are the same, up to minor corrections due to e.g. temperature changes (Supplementary Information F2). This explains the linear relation followed by the deviations within metabolic type, Eq. (7). In contrast, variations in dissipation across different metabolic types are due to large changes in the forces $r_i$, unrelated to element conservation.

This example shows how a data point is extracted from a reference. However, we stress that the type of data available in the many references of the database is highly diverse. In Supplementary Information G we describe in detail the curation process of the data.

---

the dissipative cost of growth results in a good predictor of biomass yields. Our work substantially advances these efforts by crucially distinguishing variability within and across metabolic types, and by applying a general non-equilibrium thermodynamics framework to an extended and validated dataset. For example, $r_{ed}$ generalizes the notion of "catabolic dissipation" to complex metabolic types; $r_b$ provides a definition of "anabolic free energy" not related to intracellular processes; and the thermodynamic efficiency $\eta$ is bounded by the second law for all data. Therefore, our systematic analysis reconciles many previous observations with varying degrees of support into two empirical thermodynamic rules that constrain unicellular growth across the domains of life.

Our work is inevitably affected by several limitations. First, a large part of the data we parsed lacks information on time-scales and individual fluxes. To circumvent this limitation, we quantified growth and energetics using yields instead of fluxes and defining the dissipation per electron donor, $\sigma$. Furthermore, in Supplementary Fig. 8 we show data from continuous cultures, where the rates are known, with a median dissipation rate of 1 Watt per gram of dry biomass grown. Second, in most experiments only a subset of the exchanges necessary to account for the dissipation was measured. Therefore, we relied on element conservation to estimate the missing yields, with associated

loss of accuracy. Third, our definitions of metabolic type and biomass composition were overly simplistic, as they ignored micronutrients and essential elements (such as phosphorous and sulfur). Despite being indispensable to life, we assumed that these contributions are small and would marginally affect the thermodynamic balances.

Given the importance of fluxes in cellular physiology[47] and energetics[11,48], it is crucial that future experimental studies of cellular energetics perform comprehensive measurements of all fluxes. We expect that a combination of novel experimental techniques[49,50] with coarse-grained as well as microscopic predictive models[15,25,51] will shed mechanistic insight into the origin of our empirical findings.

To conclude, our work constitutes a step towards connecting non-equilibrium thermodynamics with cellular physiology[24,47,52]. The progress we made in this direction has implications in broadly different fields. For example, the dataset can be used as the basis for models of biogeochemical cycles[53], but it can also guide research linking mammalian metabolism, growth, and thermodynamics in healthy and diseased systems[54,55]. Furthermore, as the dataset is open, it can be augmented to include other metabolic types, such as photosynthesis, thus expanding ecological studies of energetics. Finally, while we made no reference to intracellular processes, our approach can be bridged

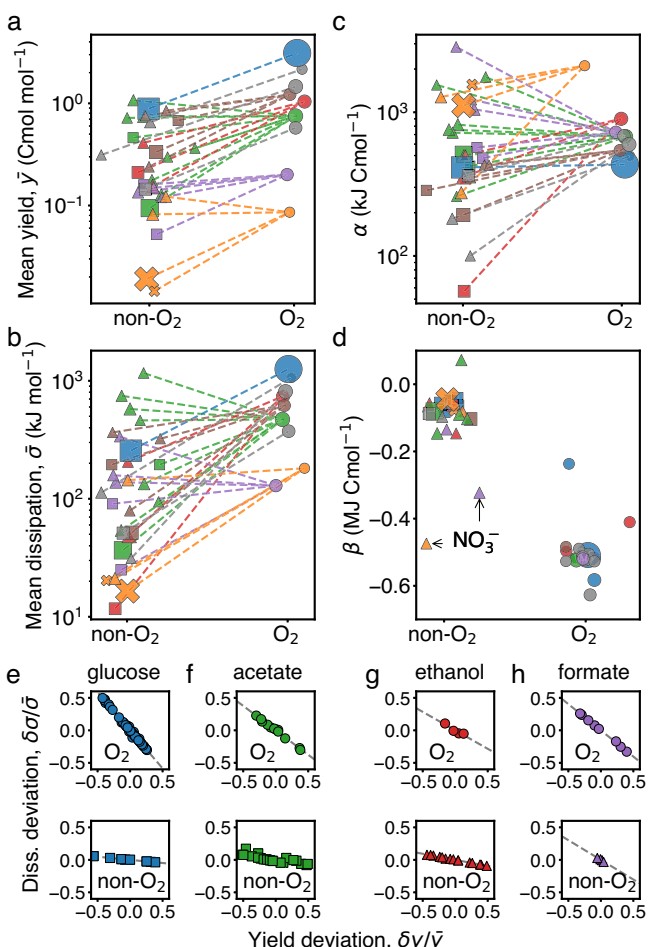

**Fig. 5 | Effect of oxygen on the dissipation-yield relation. a** Mean biomass yield, $\bar{y}$, for 30 pairs of groups of metabolic types that use the same electron donor anaerobically (non-oxygen) and aerobically (oxygen). A dashed line is drawn connecting each pair. The marker size represents the amount of data in each group. For most pairs, the aerobic group has a higher yield than the anaerobic counterpart. **b** The same as panel (**a**) but for dissipation, $\bar{\sigma}$, showing the same trend. **c** The same as panels (**a**, **b**) but for the cost, $\alpha$, which is more conserved within pairs and shows no clear trend. **d** The same as panels (**a**–**c**), but for the deviation coefficient, $\beta$. For aerobic types $\beta$ is large and negative, indicating a trade-off between yield and dissipation. Two exceptions correspond to anaerobic respiration of nitrate $NO_3^-$: reduction to ammonia (purple triangle) shows a value of $\beta$ intermediate to those of anaerobic and aerobic types, reduction to nitrogen (orange triangle) shows a value of $\beta$ indistinguishable from that of aerobic types. **e**–**h** Examples of deviation of yield and dissipation for three pairs, which agree with the trend in Fig. 3c (see Supplementary Fig. 9 for all the deviations). Panel **h** shows an exception, in which the anaerobic electron acceptor nitrate produces a steep slope similar to aerobic types.

with thermodynamics of metabolic or signaling networks[11,56]. Overall, the two empirical rules we established constitute a long sought-after connection between metabolism and thermodynamics.

## Methods

This work is based on data collected from published literature. Except for four cases, we parsed all data from the original references, even when we started from existing databases[26,28–30,57]. In Supplementary Information G1, we describe the curation process, and we provide details for each reference in Supplementary Information G3. We report the curated yields in Supplementary Data 1 and auxiliary information on the experiments in Supplementary Data 3 (see Supplementary Data 7 for all references). In Box 2, we illustrate how, starting from the experimental measurements provided in the references, we compute

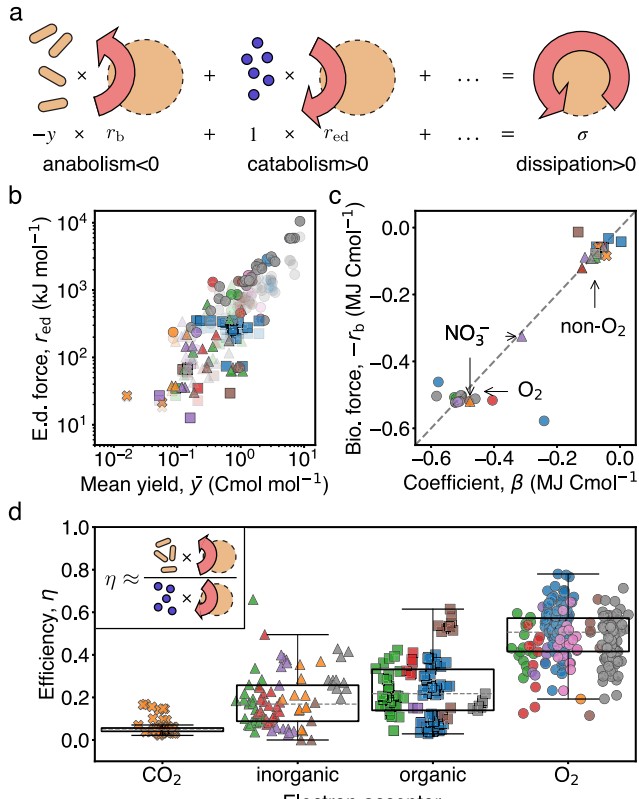

**Fig. 6 | Independent thermodynamic forces control the dissipative cost and the efficiency of growth. a** The total free energy dissipation can be decomposed in a sum of independent terms, each corresponding to the product of an independent flux with a generalized thermodynamic force. Schematic shows the dominant ones. **b** The electron donor force, $r_{ed}$, for each type is plotted against the biomass yield, $\bar{y}$, showing a strong correlation. In a transparent shade, the dissipation $\bar{\sigma}$ is plotted against the yield, as in Fig. 3b. The proximity between both shades shows that $r_{ed}$ is a good predictor for $\bar{\sigma}$. **c** The biomass force, $r_b$, for each type is plotted against the coefficient $\beta$. Both terms are essentially equal (dashed line is not a fit). A clear distinction exists between aerobic and anaerobic types, with outlier anaerobes located between them. **d** The thermodynamic efficiency of growth for all data is shown. Aerobic metabolisms are singled out by their high thermodynamic efficiency. Inset shows a pictorial definition of the efficiency as a ratio of the dissipative terms corresponding to anabolism and catabolism. The box plots are defined as in Fig. 2a. The boxes from left to right describe $n = 71, 85, 109, 239$ data points, respectively.

the dissipation of free energy. In Box 3, we provide an illustrative example. Briefly, we selected experiments compatible with the assumptions of the theoretical framework (Box 1 and Supplementary Information C). This includes controlled laboratory conditions, steady growth of a single species in continuous or batch culture, the use of minimal media, and the measurement of biomass and a minimal set of exchange fluxes or yields. Using element conservation, we compute the value of all exchange fluxes in the form of yields per electron donor (Box 2). All values are reported in Supplementary Data 2. If the elemental composition of the biomass was not reported, we used data from the closest related species available from different sources (Supplementary Information I1 and Supplementary Data 4 for the values used). The free energy dissipated per electron donor consumed is calculated using the free energies of formation of nutrients and products (Supplementary Information H), and of biomass (Supplementary Information I). We account for the temperature and pH of the experiment (Supplementary Information H), but we assume "biological standard" concentrations (1 mM) of chemical species in aqueous solution. In Supplementary Data 5, we report the thermodynamic

properties used for the exchanged chemicals. The results of the thermodynamic computations are reported in Supplementary Data 6.

**Statistics and reproducibility**
All parsed data was processed and analyzed using Python v3.8.5, with the use of NumPy, SciPy, and pandas libraries for linear algebra computations, linear fits, and statistical analysis, as explained within the main text, figure legends, and Supplementary Information.

After the initial selection of references compatible with our modeling assumptions, we excluded only a tiny fraction of the data. In a few cases, element conservation required a chemical species to be consumed, despite the reference indicating it as a product, or vice versa. This occurred with data on anaerobic fermentation in ref. 58, which we excluded from this study. In another case, five data points resulted in a negative dissipation, i.e., a violation of the second law. We excluded this source because we assumed it did not provide sufficient information on the products and substrates used in the experiment.

**Reporting summary**
Further information on research design is available in the Nature Portfolio Reporting Summary linked to this article.

## Data availability
The authors declare that the data supporting the findings of this study are available within the paper and its Supplementary Information and Supplementary Data files.

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

## Acknowledgements

All authors thank G. Falasco, R. Rao, F. Avanzini, V. Bocan, S. Pigolotti, A. Flamholz, and U. Von Stockar, as well as several members of the IGC community for feedback on the manuscript. P.S. would like to thank J. Howard and S. Leibler for insightful discussions. J.R. was supported by core funding from the Max-Planck-Society. This project has received funding from the European Research Council (ERC) under the Horizon 2020 research and innovation program (Grant agreement No. 949811 — EnBioSys) to J.R. and core funding from the Gulbenkian Foundation to P.S.

## Author contributions

All authors performed data analysis and interpretation and wrote the manuscript. T.C. parsed and collected published data. All authors conceived the project, P.S. and J.R. supervised the project.

## Funding

## Competing interests

The authors declare no competing interests.
