## [Transparent Peer Review file · Nature Communications]

Thermodynamic dissipation constrains metabolic versatility of unicellular growth

Corresponding Author: Dr Jonathan Rodenfels

Version 0:

Reviewer comments:

Reviewer #1

(Remarks to the Author)

The authors have addressed the energetics of unicellular growth, a fundamental topic in the energetics of living processes. The authors analyze the energy balance in terms of dissipation and biomass growth, finding a general trend across the variety of types of metabolism in bacteria. The general result shown in Figure 2, that an average of 500kJ/mol is dissipated per every Cmol (roughly 25 grams) of biomass yield, is beautiful. I understand that this result holds across all unicellular systems studied so far. Further on, the authors reveal a dispersion from this average result, with distinct trends in the oxidative (aerobic) and non-oxidative (anaerobic) species. The aerobic types dissipate more than the anaerobic types and, at the same time, are also more efficient (figure 5). These results are important for the ever-growing community of scientists interested in the energetics of life. It has been a pleasure to read this paper, and I am happy to recommend it for publication. However, some questions should be clarified by the authors before acceptance.

Most of all, the authors address the question of dissipation in the frame of nonequilibrium thermodynamics, synthesized in box 1. Interestingly, all quantities in equation (1,2,3) are currents or fluxes with units of enthalpy, entropy, or heat generated per unit of time. However, rates per unit time of any quantity are even mentioned in the paper. The cell colony rate of growth γ should confer to both dissipation σ and biomass yield y the quality of rates. However, the dimensionless quantity y and the dissipation σ , defined in box 1, are defined so that time dependence cancels out in the steady state. However, the entropy production rate is important (equation 3), and the authors should give estimates of this quantity across metabolic types. For example, is the average cost α (Eq.4) directly related to the average entropy production rate? My estimations from $\alpha=500\text{kJ/Cmol}$ give an average entropy production rate ($T\dot{s}_{\text{prod}}$ in Eq.3) of 6000 Watts/kg, three orders of magnitude higher than the 1Watt/kg in the human body. Is this number reasonable? This is the main part I missed in the paper, and it should be discussed.

There has been a lot of work in the field of stochastic thermodynamics and the measurement of entropy production rate. This is a topic that impacts directly on this paper. Cells are in steady states where energy consumption is used for housekeeping, maintaining them alive. At difference with the present study, the energy sources from metabolism are not used for mass production, but energy is simply released as heat to the environment. Measuring the heat rate has become a challenge in the field and this work should be outlined and some references cited, e.g. the recent measurement of the entropy production rate of individual red blood cells (Di Terlizzi et al., Science, 383.6686 (2024): 971-976) . It would be useful if the authors could also refer to existing work in the literature where \dot{q} in Eq.(3) has been directly measured for bacterial growth. Panel A in the box shows some data but again, there is plotted heat \dot{q} , but not heat per unit time.

Other questions,

1) In equation 1, the authors should discuss the relative weight of the different contributing terms. I suspect that the most important part is the heat released \dot{q} , as the biomass growth is negligible with respect to the input (substrate) and output (product) energy flows as shown in Figure 3A. On the other hand, the input and output contributions quite compensate each other (Figure 3C). These results are at the level of free energies but should also probably extend at the level of entropy flows and the term $\gamma \cdot s$ in Equation (3), making \dot{q} the dominant contribution to $T\dot{s}_{\text{prod}}$ in Eq.3.

If this is correct, then the enthalpy difference mostly contributes to \dot{q} in the Hess law balance equation (2). The authors should clarify how the low contribution from free energy storage in Figure 3 ($-\tilde{g}/\tilde{\rho}_C$) and the near

compensation between substrate and product free energies (figs 3B and 3C) translate into the relative contributions of enthalpies and entropies to generate entropy at a given rate.

In page 1 in the intro, please briefly explain what do you mean by "plastic response to environmental changes"

In panel A of the figure in the box, an endothermic metabolic type is shown as an exception to the general exothermic rule. Does this mean that this type of organisms are coolers rather than heaters? It sounds strange unless these are thermophiles living in extremely hot environments. In fact, all cells in normal organisms of the plant and animal kingdom are exothermic, as far as I know. Please clarify.

2) There are no references in the box to previous work, general refs to main results Eqs(1,2,3) should be given.

3) Is the term g in Eq.3 always positive? Please specify in the text of the box.

4) I wonder what is the reason of using units of Cmol rather than mol . I suspect this is a standard rule in the field but it should be explained, also why 1Cmol is approximately 24.5 grams (when the molecular weight is roughly 12 grams).

5) In figure 3C, is the distance separating the points above the diagonal line limiting the validity of the second law, an estimation of $\tilde{g}/\tilde{\rho}_C$? How this related to the sign of g in Eq.3?

6) In page 5, right column bottom, "three anaerobic/aerobic pairs" . Should not be four?

7) Is there any explanation why the coefficient β should be negative? Could it be positive?

8) In page 6, briefly explain the rationale and how the decomposition of the fluxes in Eq.3 into independent contributions is made.

In the section "Opposing...." in page 6, it should be explicitly shown the formula $\tilde{\sigma} = r_{ed} - r_b y$ which is never shown but helps a lot in understanding the results in figure 4.

9) In the box, it should be specified the units of $q, h, T^* s, g$. Are these energies (Joules) per unit volume? This depends on the units of f_i^+ in Eq.3 which are mol per volume and time, or Molar units per unit time.

10) I wonder how the result that aerobic unicellular organisms show large efficiencies despite the large dissipation correlates with the adaptive theory of England that biological systems tend to maximize dissipation to be more efficient. A discussion of such a theory in the frame of the current results results will be appreciated by readers.

Reviewer #2

(Remarks to the Author)

This paper compiles an impressive dataset from previous experimental studies quantifying unicellular growth, metabolism, and dissipation across a wide variety of systems. The authors develop a simple nonequilibrium thermodynamic framework to understand the relations between fluxes of chemical substrates and products and the resulting heat production and biomass growth rates. The analysis highlights a wide variation in dissipation and biomass yield (with a relatively conserved ratio) across different metabolic types and more subtle variation within metabolic type. The authors also identify distinct differences (e.g. biomass yield, efficiency, variation within metabolic type) between anaerobic and aerobic metabolism. This paper therefore provides compelling new evidence demonstrating the impact of the underlying thermodynamics on unicellular metabolic outcomes. However, a more accessible presentation along with more thorough discussion of the implications of the thermodynamic theory in the main text are necessary before publication.

Major issues:

The main text largely obscures what sort of measurements are made in the experimental datasets and how these are used with the thermodynamic theory to compute things like dissipation, yield, etc. It would be helpful (perhaps in Box 1) to provide a concrete representative example of a metabolic reaction and the measurements that feed into the calculations. For example, what is g in Eq. 3? It seems to be the free energy for biomass synthesis, but its exact definition is nowhere to be found. How is it determined (or measured)?

It is unclear how "metabolic type" is defined. It seems to depend purely on the identities of the electron donor-acceptor pair on the left hand side of the "macrochemical equation" and independent of the underlying metabolic networks. Within each "metabolic type" (MT), there are multiple datasets (experiments), which constitute the variations within the specific MT. However, the nature of these "in-type" variations is not clear. Do they correspond to variations in the underlying metabolic networks (e.g. for different organisms) or different external conditions, or both? One of the main results in this study is the linear tradeoff relation (Eq. 5) between variations of the "in-type" growth and cost. Given that these "variants" share the same donor-acceptor pair, is there a simple intuitive way to understand the linear relationship and the observation that β is negative?

In general, there needs to be a more thorough discussion of the implications of the nonequilibrium thermodynamic theory in the main text. On one hand, the theory is used as a data analysis tool, enabling computation of dissipation, yield, etc. from

the large set of experiments to demonstrate metabolic diversity across systems. However, what seems more impressive was the claim that the model predicts/explains the observed constraints on metabolic versatility (e.g. the linear relations characterized by α , β). In our opinion these results (e.g., those detailed in SI Appendix E) need to be much more thoroughly explained and emphasized in the main text. For example, it seems like the argument for the conserved cost α relies on the relationship $r_{ed} = \alpha y$ shown in Fig. 5 – is this just an empirical observation or is there a theoretical argument supporting this relation as well? Related: why does α in SI line 392 have dependence on type t ?

The fact that efficiency can be expressed in terms of variability within and between metabolic types, i.e., $\eta \sim \beta/\alpha$ (line 358), is quite surprising. Normally one would not expect the relation to other metabolic types to impact efficiency; is there an intuitive explanation for this result? For example, as efficiency, $\eta < 1$, but what is the general guarantee that this has to be the case, i.e., the proof that $|\beta| < \alpha$? Also, β is negative so there is a “-” sign missing in this expression for η . Another perhaps more natural efficiency would be growth ($\gamma * g$) divided by the total energy used/spent ($E_s = -\sigma_{sb} + \sigma_{pd}$). Here, E_s is used as the energy in the product ($-\sigma_{pd}$) is not spent but simply stored. This efficiency $\eta' = \gamma * g / E_s$ will also give the fraction of energy “wasted” in generating heat, which is simply $(1 - \eta')$.

Minor comments/typos:

In our opinion including “evolution” in the final sentence is overstating the results of the paper. While the data is suggestive, there is no direct evidence indicating that metabolic processes studied here are optimized by evolution. Along these lines, it might be useful to add a brief discussion regarding what sort of experiments/theory are necessary to more directly link metabolic properties to outcomes of evolutionary processes.

Using the approximation that efficiency $\eta = \beta/\alpha$, along with the α and β values quoted in lines 161, 198 gives an efficiency of ~ 1 for aerobic processes contradicting Fig 5. Is this simply a breakdown in the approximation?

The signs for the substrate energy and the product energy, σ_{sb} and σ_{pd} , are confusing. From their definitions given below Eq. 3, they seem to be negative? (not a good choice of convention if they are negative). But then, in Fig. 3B, labels for the x-axis seem to suggest that they are positive. Also, the x-axis label for Fig. 3A is also confusing, e.g., under the biomass, there is “ $-g/\rho_C$ ” which seems to indicate a negative growth?

Line 221 : should the figure reference be to the fourth box (rather than third)?

Reviewer #3

(Remarks to the Author)

Version 1:

Reviewer comments:

Reviewer #1

(Remarks to the Author)

The authors have done an excellent job in answering my questions. I am happy to recommend the paper for publication.
Signed: Felix Ritort

General comments for both reviewers

We thank the reviewers for their constructive comments. Addressing the comments has very substantially improved the clarity of our manuscript. A general thread through both reports is that the theoretical background and methodological approach are scattered through the SI and the box. While we aimed to improve the paper's clarity by relegating most theory to the SI, we can now see how this resulted in the opposite effect.

To address this point, we have created a theory section to help readers better appreciate the results. We would genuinely appreciate if the reviewers took the time to review this new theory section of the revised manuscript and assess whether they find the revised manuscript clearer than the previous one and point out any issues that may still be present.

Besides this general comment, we have addressed all issues raised by the referees point-by-point. Below is a response to these issues, followed by a brief description of the changes. In the revised manuscript, changes appear in blue.

Reviewer #1

The authors have addressed the energetics of unicellular growth, a fundamental topic in the energetics of living processes. The authors analyze the energy balance in terms of dissipation and biomass growth, finding a general trend across the variety of types of metabolism in bacteria. The general result shown in Figure 2, that an average of 500kJ/mol is dissipated per every Cmol (roughly 25 grams) of biomass yield, is beautiful. I understand that this result holds across all unicellular systems studied so far. Further on, the authors reveal a dispersion from this average result, with distinct trends in the oxidative (aerobic) and non-oxidative (anaerobic) species. The aerobic types dissipate more than the anaerobic types and, at the same time, are also more efficient (figure 5). These results are important for the ever-growing community of scientists interested in the energetics of life. It has been a pleasure to read this paper, and I am happy to recommend it for publication. However, some questions should be clarified by the authors before acceptance.

We thank the reviewer for their appreciation of our work and the results presented.

Most of all, the authors address the question of dissipation in the frame of nonequilibrium thermodynamics, synthesized in box 1. Interestingly, all quantities in equation (1,2,3) are currents or fluxes with units of enthalpy, entropy, or heat generated per unit of time. However, rates per unit time of any quantity are even mentioned in the paper. The cell colony rate of growth γ should confer to both dissipation σ and biomass yield y the quality of rates. However, the dimensionless quantity η and the dissipation σ , defined in box 1, are defined so that time dependence cancels out in the steady state. However, the entropy production rate is important (equation 3), and **the authors should give estimates** of this quantity across metabolic types. For example, is **the average cost α (Eq.4) directly related to the average entropy production rate?** My estimations from $\alpha=500\text{kJ/Cmol}$ give an average entropy production rate (\dot{S}_{prod} in Eq.3) of 6000 Watts/kg, three orders of magnitude higher than the 1Watt/kg in the human body. Is this number reasonable? This is the main part I missed in the paper, and it should be discussed.

Indeed, as the reviewer points out, we opted to normalize dissipation rate and biomass growth by a unit of nutrient (electron donor) consumed in this work. The primary motivation behind this choice is that the growth rate was not always reported in the data we analyzed (only in about half of the data). Therefore, to maximize the scope of our analysis, we opted to use normalized quantities. The second motivation is that using normalized quantities allowed us to compare the types of metabolism that function at very different time scales.

Nevertheless, as the reviewer points out, estimates of dissipation per unit of time should be provided. We note that in *Extended Data Fig.8*, we show the specific entropic production rate for the data in our meta-analysis that provided growth rates. From this data, the median entropy production rate per unit of biomass synthesized is 0.9 kW/kg. The reviewer remarks that the number is three orders of magnitude higher than the 1W/kg often mentioned for the human body. However, this is not a contradiction, as the problems are fundamentally different. Indeed, the estimate cited for the human body refers to the maintenance expenditure of an average adult per unit of wet mass.

In contrast, our data refers to microbial exponential growth per unit of dry mass. For example, if one considers that ribosomes hydrolyze 4ATP equivalents (100kT) per amino acid polymerized (5C atoms), this amounts to $\alpha=20kT/C$, which is a factor of four lower than our estimate. Therefore, the value we identified does not contradict the number typically reported for human maintenance dissipation, as our setup is fundamentally different.

Concerning the question of whether this number is, in fact, reasonable, we dedicated *SI D* to compare this number to other estimates from related studies. Indeed, the value of α that we found is larger, but within a 2 to 10-fold difference than the values estimated in alternative ways in other studies of dissipation of growing microbes.

Changes 1: We now motivate more clearly the reasoning behind the normalization of fluxes in the main text and in the Boxes. We also refer to the sections in the SI where dissipation per unit time is discussed, and the power dissipated from the data providing growth rates is highlighted in the discussion section.

There has been a lot of work in the field of stochastic thermodynamics and the measurement of entropy production rate. This is a topic that impacts directly on this paper. Cells are in steady states where energy consumption is used for housekeeping, maintaining them alive. At difference with the present study, the energy sources from metabolism are not used for mass production, but energy is simply released as heat to the environment. Measuring the heat rate has become a challenge in the field and this work should be outlined and some references cited, e.g. the **recent measurement of the entropy production** rate of individual red blood cells (Di Terlizzi et al., *Science*, 383.6686 (2024): 971-976). It would be useful if the authors could also refer to **existing work in the literature where \dot{q}** in Eq.(3) has been directly measured for bacterial growth. Panel A in the box shows some data but again, there is plotted heat, but not heat per unit time.

We thank the reviewer for this comment. We would like to note that our manuscript already contains many references where \dot{q} is measured, which were used to construct panel A of the box figure (now Fig.1b). However, we did not report the heat rate measured in the primary literature for the reasons explained before. We also agree with the reviewer that we should provide additional references to recent work where heat and dissipation has been measured.

Changes 2: We added a new figure, *Extended Data Fig. 8c*, where the heat flux is reported. In addition, in the discussion, we now comment on recent advances in measuring heat rates with additional references.

Other questions,

1) In equation 1, the authors should discuss the relative weight of the different contributing terms. I suspect that the most important part is the heat released \dot{q} , as the biomass growth is negligible with respect to the input (substrate) and output (product) energy flows as shown in Figure 3A. On the other hand, the input and output contributions quite compensate each other (Figure 3C). These results are at the level of free energies but should probably extend at the level of entropy flows and the term $\gamma \cdot s$ in Equation (3), making **\dot{q} the dominant contribution to $T\dot{s}_{\text{prod}}$** in Eq.3.

If this is correct, then the enthalpy difference mostly contributes to \dot{q} in the Hess law balance equation (2). The authors should clarify **how** the low contribution from free energy storage in Figure 3 ($-\tilde{g}/\tilde{\rho}_C$) and the near compensation between substrate and product free energies (figs 3B and 3C) **translate into the relative contributions of enthalpies and entropies** to generate entropy at a given rate.

We answer these two points together, as they are closely related. The reviewer raises an extremely interesting point. Given its relevance, we are working on a second paper about this topic.

Equation 1 shows that entropy is produced by exchanges of both, heat, and chemical entropy. There is no general argument to prove the prevalence of one or the other. In fact, we found that the relevance of the heat and chemical entropy terms depends on the type of metabolism. For example, the entropy production of aerobic metabolism is almost entirely due to heat production. On the contrary, in endothermic metabolism, such as acetotrophic methanogenesis shown in *Fig. Box1A Inset*, cells absorb small amounts of heat from the environment, and chemical entropy is the leading dissipative term. Finally, we note that the biomass term is usually negligible in terms of dissipation across all types of metabolism.

In principle, the compensation argument can apply similarly to the input and output of chemical entropy and chemical enthalpy (i.e. heat). Hence, compensation does not help to guess which term is more relevant.

Changes 3: We now explicitly mention our intention to investigate this point further in the Discussion.

In page 1 in the intro, please briefly explain **what do you mean by "plastic response to environmental changes"**

We thank the reviewer for pointing this out. By plastic response we mean phenotypic switching.

Changes 4: we now explain this point more carefully.

In panel A of the figure in the box, an endothermic metabolic type is shown as an exception to the general exothermic rule. Does this mean that this type of organisms are coolers rather than heaters? It sounds strange unless these are thermophiles living in extremely hot environments. In fact, all cells in normal organisms of the plant and animal kingdom are exothermic, as far as I know. Please clarify.

As the reviewer correctly suggests, these cells are coolers rather than heaters. These microbes are methanogenic archaea fermenting acetate to methane and thrive at high (but not extreme) temperatures. As mentioned above, this is a striking example of how the chemical entropy term can prevail over the heat term in the entropy balance Eq. 1. In line with the previous point, we are investigating this further in a future publication.

Changes 5: see Changes 3.

2) There are no references in the box to previous work, general refs to main results Eqs(1,2,3) should be given.

We apologize for this lack of citations.

Changes 6: we now cite the relevant literature.

3) Is the term g in Eq.3 always positive? Please specify in the text of the box.

When using free energies of formation, the sign of g , which is the free-energy content of biomass, is negative. In general, the sign depends on the reference used to express the free energies (e.g. formation or combustion).

Changes 7: (see also changes 13, 15) We now specify the sign of g using free energies of formation.

4) I wonder what is the reason of using units of $Cmol$ rather than mol . I suspect this is a standard rule in the field but it should be explained, also why $1Cmol$ is approximately 24.5 grams (when the molecular weight is roughly 12 grams).

As hinted by the reviewer, the use of $Cmol$ is standard in the field of bioengineering. The reason is that one "mole" of biomass is not well defined, since we do not resolve individual cells but rather overall biomass of the cell culture.

The molecular weight accounts for the carbon content of biomass, together with its hydrogen, oxygen, and nitrogen content. In the same way, the molecular weight of CO_2 (44 g/mol) is the sum of its carbon weight together with its oxygen weight. We would like to point to the SI Appendix F2 and H, which we dedicated to clarifying this point.

Changes 8: We now refer to the SI Appendix F2 and H when introducing biomass units in the main text.

5) In figure 3C, is the distance separating the points above the diagonal line limiting the validity of the second law, an estimation of $\tilde{g}/\tilde{\rho}_C$? How this related to the sign of g in Eq.3?

The distance from the diagonal line is indeed related to the dissipation. The closer to the line, the closer to the limit imposed by the second law. The term $\tilde{g}\tilde{\rho}_C$ offsets the diagonal line from the zero. The sign of g in Eq.3 implies that the diagonal line is below the 0 point. In other words, biomass could be considered a product, and it would contribute to σ_{pd} . In practice, this contribution is very small.

Changes 9: We now clarify this point in the figure legend of now Fig 4c (previously Fig 3C).

6) In page 5, right column bottom, "three anaerobic/aerobic pairs". Should not be four?

The sentence pointed out by the reviewer mentions three panels and three corresponding pairs. "Figures 4E-G display [...] for three anaerobic/aerobic pairs". The fourth pair, displayed in panel H, is an exception and therefore mentioned separately afterward. We acknowledge the possible confusion arising from such a presentation and thank the reviewer for pointing it out.

Changes 10: We now rephrased the text to improve clarity and made some small edits to the figures.

7) Is there any explanation why the coefficient β should be negative? Could it be positive?

To our knowledge, no thermodynamic principle requires β to be negative. Notice that β is a "macroscopic" force coupled to the overall biomass growth, and not to a molecular "microscopic" process. Yet, a negative β corresponds to the common intuition that metabolism enables growth by free energy transduction. If β and α were both positive, there would not be macroscopic free energy transduction, and the second law would not impose any macroscopic constraint on growth. Therefore, the sign of β is an important empirical result of our work.

Changes 11: We now emphasize that the sign of β is a result.

8) In page 6, briefly explain the rationale and how the decomposition of the fluxes in Eq.3 into independent contributions is made. In the section "Opposing..." in page 6, it should be explicitly shown the formula $\tilde{\sigma} = r_{ed} - r_b y$ which is never shown but helps a lot in understanding the results in figure 4.

We thank the reviewer for the suggestion.

Changes 12: we now show the formula Eq. E2 (general case of E3) and its explanation in the main text. We now also show Eq.E3 for a specific case in the Methods section.

9) In the box, it should be specified the units of q, h, T^*, g . Are these energies (Joules) per unit volume? This depends on the units of f_i^+ in Eq.3 which are mol per volume and time, or Molar units per unit time.

As correctly stated by the reviewer, these quantities are densities with units of energy per unit volume.

Changes 13: (see also changes 7, 15) We added clarifications concerning the units of these quantities where they appear, and reference the SI. Moreover, to ease the navigation of the manuscript, we added a table in the SI with units and definitions of main quantities.

10) I wonder how the result that aerobic unicellular organisms show large efficiencies despite the large dissipation correlates with the adaptive theory of England that biological systems tend to maximize dissipation to be more efficient. A discussion of such a theory in the frame of the current results results will be appreciated by readers.

We thank the reviewer for the suggestion. However, we think it is important to stress some crucial differences between England's adaptive theory and our work. The former is based on stochastic thermodynamics and focuses on the entropy production in stochastic systems described at the microscopic level. In contrast, our framework is "macroscopic", based on the theory of deterministic chemical reaction networks. The connection between stochastic microscopic and deterministic macroscopic chemical reaction networks is highly non-trivial

and a topic of ongoing research. Therefore, we prefer not to speculate on such uncertain ground and instead refer to other works more closely related to our macroscopic deterministic framework. Furthermore, given the reservations expressed by Reviewer #2 to mention evolution, we preferred not to add anything to the main discussion.

Changes 14: (see also changes 24, 25) We have removed evolutionary speculations from the discussion to also comply with comments from Reviewer #2.

Reviewer #2

This paper compiles an impressive dataset from previous experimental studies quantifying unicellular growth, metabolism, and dissipation across a wide variety of systems. The authors develop a simple nonequilibrium thermodynamic framework to understand the relations between fluxes of chemical substrates and products and the resulting heat production and biomass growth rates. The analysis highlights a wide variation in dissipation and biomass yield (with a relatively conserved ratio) across different metabolic types and more subtle variation within metabolic type. The authors also identify distinct differences (e.g. biomass yield, efficiency, variation within metabolic type) between anaerobic and aerobic metabolism. This paper therefore provides compelling new evidence demonstrating the impact of the underlying thermodynamics on unicellular metabolic outcomes. However, a more accessible presentation along with more thorough discussion of the implications of the thermodynamic theory in the main text are necessary before publication.

We thank the reviewer for their appreciation of the results presented in our work. We also greatly appreciate the honest critique of the problems that the reviewer encountered in the presentation. This manuscript has been very difficult to write and has gone over many iterations, yet we agree that the past version still lacked clarity. We have taken this criticism seriously and significantly changed our work's presentation. We would truly appreciate it if the reviewer carefully reads this revised version and expresses any further comments concerning the presentation since our goal is to have a good and clear manuscript.

Major issues:

The main text largely obscures what sort of measurements are made in the experimental datasets and how these are used with the thermodynamic theory to compute things like dissipation, yield, etc. It would be helpful (perhaps in Box1) to **provide a concrete representative example** of a metabolic reaction and the measurements that feed into the calculations. For example, **what is g in Eq. 3?** It seems to be the free energy for biomass synthesis, but its exact definition is nowhere to be found. How is it determined (or measured)?

We thank and appreciate the reviewer's suggestion on improving our presentation of the theory and its connection with the data. In the original manuscript, we relied heavily on the SI but did not reference it enough in the main text. For instance, the biomass free energy g is therein described, including a description of existing approaches to measure it as well as to provide estimates based on element composition. We apologize for the definition missing in Box 1. In retrospect, this heavy reliance on the SI resulted in a lack of clarity in the manuscript, as also pointed out by Reviewer #1. Therefore, we have opted to substantially increase the theoretical and methodological discussions in the main text, including an example, as the reviewer suggests. We hope that the revised manuscript is clearer to the reviewer.

Changes 15: (see also changes 7,13,19) We added the definition and reference to SI when g is introduced (the same for s and h) in the Methods. Moreover, we expanded the theoretical content of the main text and added an example.

It is **unclear how "metabolic type" is defined**. It seems to depend purely on the identities of the electron donor-acceptor pair on the left hand side of the "microchemical equation" and independent of the underlying metabolic networks.

We thank the reviewer for pointing out the need to clarify the definition of metabolic type. A metabolic type is defined by the set of all chemicals appearing in the "macrochemical equation", substrates (left hand side) and products (right hand side). This includes electron donor, acceptor, but also the nitrogen source as well as different products. To the extent that the substrates and products depend on the underlying metabolic network, so does the metabolic type. However, if two different networks metabolize the same set of substrates and products, they are labeled as the same metabolic type. In the figures of the main text, we only used specific

colours and symbols for the electron donor and acceptor type to avoid clutter in the figures' legend. In the SI there is a comprehensive list of all metabolic types, as well as plots in which each distinct metabolic type is depicted with a separate symbol.

Changes 16: We clarified the definition of metabolic type within the main text.

Within each “metabolic type” (MT), there are multiple datasets (experiments), which constitute the variations within the specific MT. However, the **nature of these “in-type” variations is not clear**. Do they correspond to variations in the underlying metabolic networks (e.g, for different organisms) or different external conditions, or both?

The variations within the MT depend on both, a different metabolic network and different external conditions. In some cases, this variation is due to changes in temperature that affect the growth yield. On other occasions, the changes are in the dilution rate of a chemostat, or in the concentration of nutrients in the media. Finally, in some cases, the variation is due to a different microbial species that belong to the same metabolic type. While the text mentioned this, it should have been made more explicit.

Changes 17: We note the source of this variation more explicitly through the example (changes 15).

One of the main results in this study is the linear tradeoff relation (Eq. 5) between variations of the “in-type” growth and cost. Given that these “variants” share the same donor-acceptor pair, is there a simple intuitive way to understand the linear relationship and the observation that beta is negative?

We thank the reviewer for raising this question. The linear relation largely results from element (C,H,O,N) conservation in a system with few degrees of freedom. The sign of the linear slope, i.e. the sign of beta, on the other hand, is an emergent property.

To see that the linear relation arises from element conservation, we note that the dissipation has a term that is proportional to the yield multiplied by a linear combination of chemical potentials. The best intuitive way to explain this point is to use ATP hydrolysis as an analogy, as we did in the main text. In each metabolic type, the “variants” use not only the same electron donor-acceptor pair, but the same full set of substrates and products. What distinguishes the “variant” is the amount of these metabolites that are exchanged. Due to conservation of elements, the quantities of substrates and products used by a “variant” are linearly correlated. For example, a certain metabolic type can transform the 6 carbons of glucose either into biomass or into CO₂. If a “variant” produces less biomass from those 6 carbons, it has to produce more CO₂ to dispose of those 6 atoms. Introducing these conservation laws explicitly in the expression of dissipation results in a linear relation between dissipation and yield, with the prefactor a linear combination of chemical potentials. The perturbations that generate changes in the yield can produce small changes in the chemical potentials (e.g., changes in temperature or electron donor concentration), and were not always reported in the data we analyzed. Moreover, the prefactor also depends on biomass composition, which can slightly vary in different “variants”. Therefore, an approximately linear relation is expected to arise. We hope that our improved theory section helps address this point, together with Eq.E2 (and explanation) that we moved from the SI to the Results section (changes 12).

The intuition behind the negative value of beta is as follows. At the subcellular scale, many molecular systems operate through energy transduction. In energy transduction, a chemical reaction driven by a strong negative free energy gradient is coupled to another chemical reaction in order to perform a certain biological function. Examples of systems performing energy transduction are molecular motors such as myosin or kinesin, where ATP hydrolysis couples to perform mechanical work; or the FoF1 ATP synthase, where the proton motive forces couples to the synthesis of ATP, a form of chemical work. Our finding that beta is negative suggests that the same is valid at the whole cell scale: oxidation of the electron donor releases a large amount of free energy, which is coupled to a gain in free energy associated with biomass synthesis. One may have expected this because it is known that many anabolic processes are coupled to catabolic processes that drive them.

Changes 18: (see also changes 15, 17,19) We elaborated on the sign of beta in the Results and Discussion. We expanded the theory in the Results section and in Box 3 (example) to clarify the origin of the linear relationship within metabolic types.

In general, there needs to be a **more thorough discussion of the implications of the nonequilibrium** thermodynamic theory in the main text. On one hand, the theory is used as a data analysis tool, enabling computation of dissipation, yield, etc. from the large set of experiments to demonstrate metabolic diversity across systems. However, what seems more impressive was the claim that the **model predicts/explains the observed constraints** on metabolic versatility (e.g. the linear relations characterized by alpha, beta). In our opinion these results (e.g., those detailed **in SI Appendix E**) need to be much more thoroughly explained and emphasized in the main text. For example, it seems like the argument for the conserved cost alpha relies on the relationship $r_{ed} = \alpha y$ shown in Fig. 5 – is this just an **empirical observation** or is there a theoretical argument supporting this relation as well?

We thank the reviewer for appreciating our results in light of the formalism of non-equilibrium thermodynamics. The reviewer gives a general recommendation, that there needs to be a more extensive discussion of these implications, as done in SI E. We agree with this consideration, and have made changes in this direction.

Concerning the predictive potential of the nonequilibrium formalism, we would like to emphasize that since our framework is basically model-free, it is not predictive. Instead, our findings are best understood as empirical observations after a “thermodynamically judicious” choice of variables. The linear relation within types is a direct consequence of element conservation, which is used as a data analysis tool. So it is not a result, but rather a fundamental assumption enforced on the data. Instead, the result is the particular value of the slope, and how it changes depending on whether the metabolic type is aerobic or not. The linear relationship across types is a purely empirical observation, and one of our two main results. We find that this linear relationship is primarily driven by the correlation between r_{ed} and yield, but we do not have a theoretical argument to explain this finding. Developing such theoretical arguments is a next step in our research program.

Changes 19: (see also changes 12,15,18) Given the feedback of both reviewers, we have substantially expanded the theoretical considerations in the main text. Importantly, we now show a key equation that used to be in SI E in the Results of the main text. Furthermore, we provide a detailed example in Box 3 of how to apply the formalism of nonequilibrium thermodynamics to data, and how element conservations underlie the linear relation observed within-metabolic type. We have also added a comment in the discussion about the need to move towards a theoretical model that explains the empirical observations of this work. These changes have significantly improved the clarity of our work and thank both reviewers for their feedback.

Related: why does alpha in SI line 392 have dependence on type t?

We define alpha as the dissipation per unit biomass for each metabolic type. Therefore, there is one alpha for each metabolic type (hence the dependence on t), which allows us to show a histogram of the different alpha values as an inset in current Fig. 4a (previously 3A). In the main text, we dropped this explicit dependency to ease the notation.

Changes 20: When we define alpha in the main text, we now explicitly clarify its dependence on the metabolic type.

The fact that efficiency can be expressed in terms of variability within and between metabolic types, i.e., $\eta \sim \beta/\alpha$ (line358), is quite surprising. Normally one would not expect the relation to other metabolic types to impact efficiency; is there an intuitive explanation for this result? For example, as efficiency, $\eta < 1$, but what is the general guarantee that this has to be the case, i.e., the proof that $\beta < \alpha$?

As we elaborate also below, the relation between efficiency and (α, β) is related to the conservation of elements, see also remarks above. By considering element conservation explicitly, we were able to identify the generalized chemical forces that couple to independent fluxes, r_{ed} and r_b . What is striking is that r_{ed} (which is a quantity involving only stoichiometry and free-energies of chemicals) correlates well with the biomass yield y . In other words, r_{ed} is the dominant contribution to the dissipation σ . Therefore, we can (roughly) approximate $r_{ed} \approx \alpha y$, which is behind the expression $\eta \approx \beta/\alpha$. Since alpha varies little across types, and beta has two distinctive values for aerobic and anaerobic types, aerobic types are more efficient.

The proof that $\eta < 1$ is a direct consequence of applying the second law of thermodynamics to the definition of α . Positive contributions to the dissipation are placed in the denominator of η , while negative contributions are placed in the numerator of the η . Since dissipation is bound by the second law to be positive, it follows that $\eta < 1$. In general, there is no proof that $|\beta| < \alpha$, but it happens to be true for most of the data points. However, this would not prove a bound on η , because $\eta \approx \beta/\alpha$ is only a rough approximation (see also below).

Changes 21: We rephrased the explanation of the relation between the variability of efficiency and within/across variability. We hope that this clarifies the independence of the efficiency on other metabolic types.

Also, β is negative so there is a “-“ sign missing in this expression for η .

We thank the reviewer for pointing out the missing sign.

Changes 22: We corrected this typo.

Another perhaps more natural efficiency would be growth (γ_g) divided by the total energy used/spent ($E_s = -\sigma_{sb} + \sigma_{pd}$). Here, E_s is used as the energy in the product ($-\sigma_{pd}$) is not spent but simply stored. This efficiency $\eta' = \gamma_g/E_s$ will also give the fraction of energy “wasted” in generating heat, which is simply $(1-\eta')$.

The reviewer raises a topic that has been widely debated in the literature (see e.g. [2]). As the reviewer points out, there are multiple possible definitions of efficiency. The definition that we used is very similar in spirit to the one proposed by the reviewer. However, before separating the fluxes of “what is gained” and “what is dissipated”, we took into account the constraints on the fluxes. This implies that our definition of efficiency has a simple relation to the measured yield (it is essentially proportional to it), and that it truly compares the fluxes coupled to their net forces. Nevertheless, we agree that the definition of efficiency proposed by the reviewer is intuitive, as others in the literature. By its own nature, an efficiency is defined to be instrumental to some application of interest. Ultimately, the convenience of our definition of efficiency is that it combines the two main results of our work into one quantity.

We would like to mention that, since we will provide the data as supplementary files, it will be easy for any reader to compute alternative definitions of efficiency. For completeness, we report below a plot of the efficiency proposed by the reviewer in the range [0,1.5].

Changes 23: We now comment in the main text on the existence of different definitions of efficiency, and on the convenience of our choice.

Minor comments/typos:

In our opinion including “evolution” in the final sentence is overstating the results of the paper. While the data is suggestive, there is no direct evidence indicating that metabolic processes studied here are optimized by evolution.

We agree with the reviewer that the connection between the results of the manuscript and evolution are not direct, but only speculative. We acknowledge that to speculate on the evolutionary optimization of the metabolic processes studied would require a dedicated discussion, rather than a passing sentence, as was in the original manuscript.

Changes 24: (see also changes 14, 25) We removed references to evolution from the last sentence.

Along these lines, it might be useful to add a brief discussion regarding what sort of experiments/theory are necessary to more directly link metabolic properties to outcomes of evolutionary processes.

We thank the reviewer for this suggestion. In a similar line, Reviewer #1 asked us to speculate about an existing theory on the thermodynamics of evolution. As the paper is already quite long, and made longer by theory additions, we have decided to remove comments about evolution. We hope the reviewer finds this solution satisfactory.

Changes 25: (see also changes 14, 24) We removed references to evolution from the discussion.

Using the approximation that efficiency $\eta = \beta/\alpha$, along with the alpha and beta values quoted in lines 161,198 gives an efficiency of ~ 1 for aerobic processes contradicting Fig 5. Is this simply a breakdown in the approximation?

As correctly suggested by the reviewer, this is a limit of the approximation. Such breakdown shows the roughness of the approximation, which amounts to the approximation $0.6 \approx 1$.

Changes 26: We now refer to Fig.6b and c which show the quality of the approximations used to write $\eta = \beta/\alpha$.

The signs for the substrate energy and the product energy, σ_{sb} and σ_{pd} , are confusing. From their definitions given below Eq. 3, they seem to be negative? (not a good choice of convention if they are negative). But then, in Fig. 3B, labels for the x-axis seem to suggest that they are positive.

We agree with the reviewer that a good choice of convention is to have positive σ_{sb} and σ_{pd} . Indeed, according to our definition this is the case (not in all of our datapoints, but in 501 out of 504), as the reviewer inferred from previous Fig. 3B. We believe that the confusion arises from the sign of the free energies of formations of chemical species, i.e the μ_i in Eq.3 and definitions below. Most, but not all, free energies of formation are negative (45 out of 54 chemicals considered), hence the minus in the definitions of σ_{sb} and σ_{pd} .

Changes 27: (see also changes 7) We now explicitly state that the majority of free-energies of formation are negative, making σ_{sb} and σ_{pd} positive.

Also, the x-axis label for Fig. 3A is also confusing, e.g., under the biomass, there is " $-g/\rho_C$ " which seems to indicate a negative growth?

The free energy of formation of biomass is negative, $g < 0$. We believe that this creates the confusion mentioned by the reviewer, as in the point above, and the wrong impression of a negative growth.

Changes 28: (see also changes 7) We now clarify in the main text the sign of the free energy of formation of biomass, to avoid confusion (also in response to Reviewer #1, changes 7).

Line 221: should the figure reference be to the fourth box (rather than third)?

We thank the reviewer for pointing out the imprecision. The precise reference is to the fourth box, or third term on the r.h.s. of the equation in the x-axis label.

Changes 29: we now refer to the fourth box.
OK

References used in this response:

[1] T.M. Hoehler, D.J. Mankel, P.R. Girguis, T.M. McCollom, N.Y. Kiang, B.B. Jørgensen, The metabolic rate of the biosphere and its components, *Proc. Natl. Acad. Sci.*, 120 (25) e2303764120, (2023).

[2] Heijnen, J. J., and J. P. Van Dijken. "In search of a thermodynamic description of biomass yields for the chemotrophic growth of microorganisms." *Biotechnology and Bioengineering* 39.8 (1992): 833-858.